# *Meizi*-Consuming Culture That Fostered the Sustainable Use of Plum Resources in Dali of China: An Ethnobotanical Study

**DOI:** 10.3390/biology11060832

**Published:** 2022-05-28

**Authors:** Yanxiao Fan, Zhuo Cheng, Qing Zhang, Yong Xiong, Bingcong Li, Xiaoping Lu, Liu He, Xia Jiang, Qi Tan, Chunlin Long

**Affiliations:** 1Key Laboratory of Ecology and Environment in Minority Areas (Minzu University of China), National Ethnic Affairs Commission, Beijing 100081, China; 19400214@muc.edu.cn (Y.F.); 18515952092@163.com (Z.C.); zhangqing9783@126.com (Q.Z.); xiongyong@ynni.edu.cn (Y.X.); leebingcong@foxmail.com (B.L.); luxiaop_1881@163.com (X.L.); helen1999331@163.com (L.H.); jjjiangxia3645@163.com (X.J.); 2College of Life and Environmental Sciences, Minzu University of China, Beijing 100081, China; 3School of Ethnomedicine & Ethnopharmacy, Yunnan Minzu University, Kunming 650500, China; 4School of Landscape Art, Yunnan Agricultural University, Kunming 650201, China; tanqi0992@163.com; 5Key Laboratory of Ethnomedicine (Minzu University of China), Ministry of Education, Beijing 100081, China; 6Institute for National Security Studies, Minzu University of China, Beijing 100081, China

**Keywords:** *Prunus mume*, Bai people, ethnobotanical methods, *Meizi*-consuming culture, sustainable use

## Abstract

**Simple Summary:**

The Bai people living in Eryuan of Dali, Yunnan, China have a long history of consuming the fruits of *Prunus mume*. As a result, the locals have developed a meaningful *Meizi*-consuming culture, which is of great significance to the conservation of local plum resources and local economic development. This study attempts to explain the relationship among the Bai’s traditional *Meizi*-consuming culture, conservation of local plum resources, and development of the plum industry in Eryuan. The ethnobotanical investigation showed that the local plum industry, which relies on the traditional Bai *Meizi*-consuming culture, improved the livelihood of Bai people and promoted local economic development. Almost every family of the Bai cultivates plum trees in their yards and spontaneously protects the local environment, so as to maintain a good ecological environment for wild plum genetic resources. With the sustainable use of plum resources, Eryuan’s plum development has created a win–win model combining traditional culture with biodiversity conservation.

**Abstract:**

*P**runus mume* has been cultivated for more than three millennia with important edible, ornamental, and medicinal value. Due to its sour taste, the *Prunus mume* fruit (called *Meizi* in Chinese and *Ume* in Japanese) is not very popular compared to other fruits. It is, however, a very favorite food for the Bai people living in Eryuan County, Dali of Yunnan, China. The local people are masters of making various local products with plum in different ways. In this research, we conducted field investigations in Eryuan County using ethnobotanical methods from August 2019 to July 2021, focusing on the *Prunus mume* (for its edible fruits). A total of 76 key informants participated in our semi-structured interviews. The survey recorded 37 species (and varieties) belonging to 11 families related to the Bai people’s *Meizi*-consuming culture. Among them, there are 14 taxa of plum resources, including one original species and 13 varieties. These 37 species are either used as substitutes for plum due to their similar taste or as seasonings to improve the sour taste of plum. The higher Cultural Food Significance Index value implies that *Prunus mume*, *Chaenomeles speciosa*, *Phyllanthus emblica*, *Prunus salicina*, and *Chaenomeles cathayensis* have high acceptance and use value in the Bai communities. Among the various local products traditionally made by the Bai people, carved plums, preserved plums, perilla-wrapped plums, and stewed plums are the most famous and popular categories in the traditional markets. Currently, the plum business based on the traditional *Meizi*-consuming culture of the Bai people is already one of Eryuan’s economic pillars. This study showed that plums play an important role in expressing the local cultural diversity, and they also help the local people by improving their livelihood through their edible value. In turn, for the sustainable use of plum resources, the Bai people positively manage local forests through a series of measures to protect the diversity of plum resources and related plant communities.

## 1. Introduction

The diversity of local products or geographical indication (GI) products is ubiquitous, as can be seen in the abundance of beverages, cheeses, meats, oils, pastries, fruits, and vegetables [1,2]. Most of the time, local products are associated with biocultural diversity. Whether they are raw materials or processed bioresources, local products are directly related to biological processes when they are cultivated, reared, and produced [3,4,5]. Certain local products are based on complex ecosystems that maintain various forms of biodiversity (from landscapes to microbial ecosystems) and include multiple plant species and animal species [6,7]. However, among the factors that affect the biodiversity, the knowledge forms and craft practices associated with local products are the most obvious because they are most directly visible [8,9]. Today, it has become a consensus that traditional knowledge related to local products or geographical indication (GI) products is of great significance for biodiversity conservation [10].

*Prunus mume* Sieb. & Zucc. (Rosaceae) has different names, such as plum, Chinese plum, Japanese apricot, *mume* plum, *mei* plum, and *mei* flower. The edible fruit is commonly called *Meizi* in Chinese or *Ume* in Japanese. This small arbor species originated from China [11,12,13]. Now, plum can be found almost all over the world [14,15,16]. According to literature reports, the application history of plum is more than 3000 years in China [17,18]. Chinese people value the ornamental and cultural purposes of *mei* flower (ornamental forms of *Prunus mume*), while they adore the edible and medicinal value of *mume* plum or *Meizi* (for harvesting fruit). For example, there are many ancient poems praising the *mei* flower in almost all dynasties, representing many good characteristics in China [19,20,21]. However, for most people, the edible and medicinal value of *mume* plum arouses their interest more than the ornamental and cultural value of *mei* flower.

In the early Shang and Zhou dynasties (about 1600–256 B.C.), Chinese people had already started to treat plum fruit as a condiment as important as salt [22,23]. The tradition of consuming plum fruit has continued. *The Romance of the Three Kingdoms* recorded the allusion of quenching one’s thirst upon thinking of plum fruit in the Eastern Han Dynasty (25–220 A.D.) [24]. During the Tang Dynasty (618–907 A.D.), plum fruit was processed into candied plum, being considered as something good to pay tribute to the emperor [25]. The Japanese plum-consuming culture was spread by Chinese monks who traveled eastward [26,27]. In the Song Dynasty (960–1276 A.D.), after the central government set up the Candied Fruit Bureau for making preserves and snacks, preserved plum gradually emerged, and it was said to be a delicious delicacy used by storytellers to quench their thirst [28]. The so-called “sour plum drink”, which has been passed down to the present, was innovated and became a popular delicacy among folk in the Qing Dynasty (1636–1912 A.D.) [29].

*Meizi* also has a long history of medicinal use in China. Its medicinal efficacy was first recorded in *Shanghan Zabing Lun,* one of the works of traditional Chinese medicine, which pointed out that plum can relieve cough, diarrhea, pain, hemostasis, and thirst [30]. Subsequently, *Shennong’s Classic of Materia Medica*, a classic of traditional Chinese medicine, also recorded that plum can reduce heat and dryness, treat physical pain, and be used for beauty and skin care [31]. *The Compendium of Materia Medica*, known as the ancient encyclopedia of China, also mentioned that plum can strengthen the stomach, moisten the lungs, warm the spleen, stop bleeding and phlegm, reduce swelling and detoxification, generate saliva, and quench thirst [32].

In addition, the ornamental and cultural value of *mei* flower also attracts people’s attention [33,34]. Ornamental plums are suitable for both potting and gardening. Chinese people have known how to appreciate *mei* flower since ancient times [35,36,37,38]. The first monograph on *mei* flower “*Fan Cun Mei Pu*” was issued during the Song Dynasty [39].

Because of its outstanding edible and medicinal functions and irreplaceable ornamental values, plum has spread to almost all parts of the world through culture exchanges and international trade between China and the world. However, when it comes to the importance of *Meizi*-consuming culture, no one can compare with the Bai people in Dali Bai Autonomous Prefecture, Northwest Yunnan, China.

With a population of 1.93 million, the Bai ethnic group is the 15th largest minority in China. They are mainly distributed in the Dali Bai Autonomous Prefecture in Yunnan Province. Dali Prefecture is the origin and main settlement of the Bai people. About 80% of the Bai people in China live here. The Bai people have their own language that belongs to the Sino-Tibetan family. They are good at artistic creation, and their architecture, sculptures, and paintings are full of their own characteristics. The economy of the Bai people is dominated by agriculture, supplemented by handicrafts and commerce. Their production level is nearly the same as that of the surrounding Han people [40]. The Bai people believe in animism. They believe that mountains, rivers, trees, insects, fish, birds, and beasts are parts of nature and should not be destroyed. They traditionally eat sour and spicy food, especially some wild fruits with sour taste. Because they love eating plums, the Bai people living in Dali Prefecture have formed a unique *Meizi*-consuming culture through continuous practice and exploration. Dried plum, plum sauce, and plum vinegar are the necessary condiments for Bai people to make delicious food. Fresh plum, stewed plum, and carved plum are snacks in the Bai people’s daily lives. In the Bai communities, there is a popular proverb that *eating apricots results in sickness but eating plum will prolong life*.

The *Meizi*-consuming culture in Eryuan is the most traditional and typical among all Bai communities. The local people here are able to earn a livelihood by processing and selling local products related to plums. Thus, Eryuan is regarded as the center of Bai’s *Meizi*-consuming culture. The plum products from Eryuan were approved on 28 December 2007 by the General Administration of Quality Supervision, Inspection, and Quarantine of China to implement GI product protection because of their good quality. Unfortunately, there is no scientific and comprehensive report about the traditional *Meizi*-consuming culture (TMCC) of Bai people in Eryuan. Therefore, in this study, our aims were (1) to investigate the traditional knowledge related to plum in the Bai community, (2) to record the relevant plant information ethnobotanically, (3) to analyze the importance of the *Meizi*-consuming culture of related plants using a quantitative evaluation method, and (4) to evaluate the relationship between plum-associated traditional knowledge and local plum resources conservation.

## 2. Methods

### 2.1. Study Area

Eryuan is one of the counties in Dali Bai Autonomous Prefecture of Yunnan Province (99°94′ E, 26°10′ N). It is located in the north of Dali City and covers an area of 2875 km^2^ (Figure 1). There are many linguistic groups in this area, including Han, Bai, Yi, Lisu, Naxi, Dai, and Tibetan, among which the Bai features the largest population. Eryuan belongs to the northern subtropical plateau climate; it is warm in spring, dry in winter, and rainy in summer and autumn. However, the diverse topography also makes the local three-dimensional climate and regional microclimate obvious. The annual average temperature in Eryuan is about 15 °C, the annual average rainfall is about 750 mm, and the annual sunshine hours are nearly 2500 h. The annual effective accumulated temperature is 4100 °C. In addition, the soil types of Eryuan are diverse, and the soil layer is deep and fertile. The climate, rainfall, sunshine, and soil texture of Eryuan are suitable for the growth of plum trees. Eryuan County has been recognized as the plum base by the Yunnan provincial government since 1987 [41].

### 2.2. Ethnobotanical Survey

The ethnobotanical field investigations were carried out in Eryuan County from August 2019 to November 2021. We chose four different types of destinations for conducting interviews, including government agencies (Forestry and Grassland Administration of Eryuan, FGA; Agriculture and Rural Affairs Bureau of Eryuan, ARA), companies (Yunnan Dali Erbao Industrial Co., Ltd., EB), traditional markets (Yuhu farmers market, YH), and six villages (Liancheng, LC; Qiaohou, QH; Datong, DT; Jiangdeng, JD; Laping, LP; and Taiping, TP). The research site is shown in Figure 1. Before the interviews during investigations, each informant who participated was informed of the purpose of the project, and their consent was obtained. Data were collected through free listing, semi-structured interviews, and participatory observation [42,43]. A total of 76 key informants were selected using the snowball sampling method, of whom 23 were male and 53 were female [44]. Their average age was 47 years old. The periodical markets in Eryuan County and Dali City were also visited [45].

Semi-structured interviews were conducted concerning the questions listed in Table 1. During this process, voucher specimens were collected. The nomenclature of all vascular plants follows *Flora of China* (www.iplant.cn/foc, accessed on 17–25 August 2020) and World Flora Online (www.worldfloraonline.org, accessed on 23–28 August 2020) [35]. Prof. Chunlin Long identified the plant species, and the voucher specimens were deposited in the herbarium at Minzu University of China, in Beijing.

### 2.3. Data Analysis

The Cultural Food Significance Index (CFSI) was calculated to evaluate the cultural significance of plants related to the traditional knowledge about *Meizi* consumption [46,47]. The calculation formula is as follows:CFSI = FQI × AI × FUI × PUI × MFFI × TSAI × FMRI × 10^−2^,
where FQI (frequency of quotation index) is the number of people who mentioned a certain plant species among all information reporters, AI is the availability index, FUI is the frequency of utilization index, PUI is the parts used index, MFFI is the multifunctional food utilization index, TSAI is the taste score appreciation index, and FMRI is the food-medicated role index. A higher CFSI value of a species indicates a more important role in the lives of local people. The classification and assignment of each index are shown in Table 2.

## 3. Results and Discussion

The study area retains a wealth of traditional knowledge related to plums, including the cultivation and processing of plums. The Eryuan plum industry based on the Bai’s TMCC not only increases the income of Bai people and promotes local economic development, but also improve the conservation and maintenance of local environment for growing plum and other organisms.

### 3.1. Plant Diversity of TMCC

According to the survey, a total of 37 plant taxa belonging to 11 families related to the traditional Bai *Meizi*-consuming culture were identified (Table 3). They are either used as raw materials, as seasonings, or as substitutes because they taste similar to plums. Rosaceae was found to be the largest family with 25 species (and intraspecific taxa), followed by Rutaceae with three species, Solanaceae with two species, and other families (Elaeagnaceae, Fabaceae, Lamiaceae, Lauraceae, Myricaceae, Myrtaceae, Phyllanthaceae, and Schisandraceae) with one species each. Within Rosaceae, 14 species (taxa) were plums (*Prunus*), including one original species and 13 varieties. The used parts of these 37 species (taxa) are also diverse, including the fruit, root, leaf, bark, bud, seed, flower, and peel, among which the most commonly used parts are the fruits (78%), followed by the seeds (5%).

The plants used by Bai people for TMCC include condiments, vegetables, food for therapy, fruits, snacks, and beverages through direct consumption, as well as cold dishes, seasoning, water boiling, processing with honey, soaking with liquor, and stewing. Thirty-five species were found to be of medicinal value. They are used to treat various disorders and diseases including thirst, digestion problem, stomach pain, inflammation, phlegm, cold and headache, cough, sore throat, weakness, blood circulation ailments, dry lungs, and rheumatism.

### 3.2. The CFSI Value of Plants Used in TMCC

The Cultural Food Significance Index (CFSI) of total plants was calculated, as shown in Table 4. A total of five species (*Prunus mume*, *Chaenomeles speciosa*, *Phyllanthus emblica*, *Prunus salicina*, and *Chaenomeles cathayensis*) had CFSI values above 2500, indicating that these species have high acceptance and high utilization value in the Bai community and play an important role in the daily life of the Bai people. The CFSI values of *Prunus mume* var. *macrocarpa*, *Glycyrrhiza uralensis*, *Cinnamomum cassia*, *Malus prunifolia*, and *Syzygium aromaticum* were below 50, while the value of *S. aromaticum* was only 3.64. However, this does not mean that the use values of these species are low, only that they are not as accepted as *P. mume*, *C. speciosa*, or other species in the Bai community, while they may be of great significance in other communities or fields [48,49,50,51,52,53].

### 3.3. Plum Cultivation, Management, and Picking

In Eryuan, almost every household grows plums in their yards or in front of and behind the houses, for the convenience of picking in need (Figure 2). The Bai people also have their own unique views on planting plums.

According to the information from key informants, plum planting is generally carried out in the rainy spring and summer. Seedlings with strong branches, full axillary buds, and intact roots are selected and planted in a place with enough sunshine, fertile soil, good drainage, and convenient irrigation. If it is a plum plantation, the planting density is controlled. After 1–3 years of growing, dwarf crops such as vegetables and beans or green manure plants can be intercropped in the orchard, which can be beneficial for this agricultural ecosystem. According to the identification, green manure plants grown by the local people are usually *Vicia sativa* L., *Astragalus sinicus* L., and *Trifolium repens* L. According to key informants, they enhance the fertility of soil when grown under the forests. The locals also use these plants as feed for poultry and livestock.

In addition, during the planting period of plums, attention should be paid to the soil, fertilizer and water management, training and pruning, and pest and disease control. For example, there are many diseases and insect pests on plum trees after the rainy season. The local people prune infected branches, improve the ventilation and light transmission conditions, and ditch the drainage to control humidity, so as to strengthen the tree and improve its disease resistance. The locals also use several chemical pesticides or biological pesticides with high efficiency, low toxicity, and low residue to control some special pests that are hard to kill.

The harvesting period of plums mainly occurs from June to July each year. Some varieties, such as *Prunus mume* var. *bung*, *Prunus mume* var. *cernua*, and *Prunus mume* var. *alplandii* mature in May. Fruits with golden color and full shape are ready for harvesting.

### 3.4. The Processing of Four Famous Plum Delicacies

The Bai people in Eryuan have their own knowledge and experience in processing plums into various local products, such as carved plums, soaked plums, and crispy plums. They can also be added with other plant condiments to make preserved plums, perilla-wrapped plums, and stewed plums. In Eryuan, the most famous plum products are carved plums, preserved plums, perilla-wrapped plums, and stewed plums (Figure 3).

Carved plums feature carvings on the plum surface with a flower-like design. The raw materials for carving plums are usually the fruits of *Prunus mume*. The fruits must be plump and golden with firm flesh, so that the carved shape looks good. Before carving, plums are soaked in salty water or lime water to make them soft and easy to carve. Next, a special plum carving knife is used to draw lines around the plum fruit; then, the fruit is gently pressed with the tip of the knife along the lines, and the plum kernel is removed while maintaining the plum structure. With a gentle press, the plum looks like a golden chrysanthemum. There is a saying in the Bai communities that a Bai bride presents carved plums made by herself to her mother-in-law as the first gift. The quality of carved plums is treated as a potential criterion for her new family to infer whether she is virtuous and capable.

Preserved plums are made following four steps: pickling of plum fruit, soaking liquid preparation, plum fruit soaking, and air drying. Firstly, the ripe fruit of *P**. mume*, *P. mume* var. *cernua*, or *P**. mume* var. *pallescens* is dried in the shade to soften and dehydrate it, and then the softened fruit is soaked in salt water for about 4 h; this step is repeated three times for pickling. Next, dried *Glycyrrhiza uralensis* roots, *Cinnamomum cassia* barks, *Syzygium aromaticum* buds, *Illicium verum* fruits, and other condiments are boiled and filtered in a certain proportion to make the liquid for soaking plum fruit. Then, the pickled plum fruit is immersed into the liquid and rotated slowly to fully absorb the liquid. Finally, preserved plums can be obtained after air-drying. Preserved plums are salty, sour, sweet, and fragrant, and they can be used as seasonings and snacks to generate saliva and quench thirst.

Perilla-wrapped plums, as described, are marinated plums wrapped with *Perilla frutescens* leaves. The mature fruits of *P. mume* var. *bung* and *P. mume* var. *cernua* are usually used to make perilla-wrapped plums. To make perilla-wrapped plums, the plums soaked in brine are drained, with the pits removed, and then thoroughly mixed with brown sugar, honey, peppercorns, and other condiments. Next, each plum is wrapped with a piece of perilla leaf and placed evenly in a clay pot. Finally, an appropriate amount of Baijiu is sprinkled in the pot for fermentation. The marinated perilla-wrapped plums can be stored for many years with unchanged taste and certain medicinal value.

Stewed plums are made by putting plum fruit into a sand pot, adding salt and *Z. bungeanum* fruits to cover them tightly, placing them in the middle of the fireplace, stacking rice husks around them, lighting the fireplace, and stewing with a low fire for 1 to 2 days. The black stewed plum is extremely sour and fragrant. It is an essential condiment for Bai people to make traditional delicacies, such as cold Dali rawhide (a traditional Bai dish made from pork skin) and hot and sour fish. In addition, the Bai people also use stewed plum with sugar and water as a drink to relieve heat in the summer.

### 3.5. Industrialization of Plum Products

In addition to the above four well-known local plum products, we found others such as soaked plums, crispy plums, rose-like plums, wolfberry-like plums, pearl plums, dried tangerine peel plums, dried plums, plum jam, plum vinegar, and plum wine. Moreover, the locals also process non-plum fruits with similar plum tastes using methods for making plum delicacies, such as *Phyllanthus emblica*, *Chaenomeles cathayensis*, and *Malus prunifolia* (Figure 4). In Eryuan, a dazzling array of the local plum products is displayed at the traditional markets.

According to the data provided by the Eryuan County Forestry and Grassland Bureau and the Agriculture and Rural Bureau, Eryuan took the opportunity of the applied protection of Eryuan plum GI products to increase integration efforts in 2006, expanding the county’s plum plantation area to 5534 hm^2^, with an output of 10,500 tons and an output value of 67.5 million RMB (CNY). In 2009, Eryuan increased the cultivation area of plums to 6300 hm^2^, and the yield of plums was 11,100 tons. In 2012, the total output of *Meizi* (plum fruit) in Eryuan reached more than 12,000 tons, and the income of plum farmers was more than 20 million CNY. In 2015, the Eryuan County Government actively guided farmers to vigorously plant plums in suitable areas, led township- and village-level governments to set up planting demonstration models, and built high-quality seedling nursery bases. Numerous high-quality fruit plum varieties, such as *Prunus mume* var. *pendula*, *Prunus mume* var. *cernua*, and *Prunus mume* f. *viridicalyx*, have been cultivated. In 2018, the county built a plum fruit production base of more than 8000 hm^2^, with an industrial output value of 360 million yuan and a plum farmer’s income of 100 million CNY, which comprehensively improved the overall quality and efficiency of plum industrialization. As the main source of income for local farmers, plum is an important commodity for people in mountainous areas to alleviate poverty.

At present, Eryuan has established more than 20 large-scale enterprises producing and processing fruit and plum products, as well as more than 350 family-based workshops of initial processing, providing more than 5000 jobs. These enterprises have developed four categories with more than 120 varieties of preserved fruits, fruit wine, beverages, and condiments using plums as raw materials. In addition to covering the markets of various cities (prefectures) in the province, the products are also marketed in Beijing, Tianjin, Shanghai, Changsha, and other metropolises, as well as exported to Japan, South Korea, and other countries. The county’s plum industry has formed an integrated pattern of production, processing, and trade. Its industrial chain development model of “company + base + farmer” is becoming more and more complete.

The development of the plum industry of the Bai people in Dali relies on the rich local plum genetic resources and the plum production technology that the Bai people continue to explore, and it also benefits from the traditional knowledge accumulated and passed down by the local people from generation to generation. Today, the plum industry is already the economic mainstay in Eryuan and one of the main income sources in other Bai areas. It is still an important and representative part of local food culture.

### 3.6. Relationship between TMCC and Plum Resources Conservation

Research shows that indigenous areas are often hotspots rich in biodiversity and cultural diversity [54,55,56]. On the one hand, in indigenous communities, organisms play an active role in expressing indigenous cultural diversity such as tools, food, medicine, and building materials. On the other hand, the indigenous culture formed around native organisms, such as traditional ecological concepts and village rules and regulations, restricts local people from damaging the environment and endangering local organisms [57,58]. The biodiversity of indigenous areas is positively correlated with the cultural diversity of indigenous peoples [59,60,61,62].

Eryuan’s geographical and climatic conditions have nurtured the local rich plum resources and biodiversity, which has laid a solid foundation for the emergence, inheritance, and development for the Bai people’s TMCC. Various local plum products based on the TMCC have improved the livelihood of the Bai people and boosted the local economy. For the sustainable use of plum resources, in the development of the plum industry, Eryuan attaches great importance to the ecological benefits brought by the industry, actively developing and expanding the economic benefits of the plum industry in accordance with the requirements of the ecologicalization of industrial development and the industrialization of ecological construction. This local characteristic industry has become a major advantage in terms of policy implementation called greening barren mountains in Eryuan to maintain ecological balance and increase farmers’ income. According to the statistics of the local government, there are nine townships, 90 villages, and more than 130,000 households growing plums in the county. There are 2211 households with more than 300 plum trees, 60 households with more than 500 plum trees, and 29 households with more than 1000 plum trees. In addition, there are 3393 pieces of plum forest with more than 50 plum trees, involving as many as 14 varieties of plums. The plantation area of plums in the *Meizi* base has reached more than 8000 hm^2^. These measures are of great significance for the protection of the local ecological environment and the conservation of the diversity of plum genetic resources.

For the wild plum resources in the mountains and forests, Bai people positively protect the local biodiversity and environment by formulating corresponding village rules and developing traditional ecological concepts established by conventions, so as to maintain a better growing environment and survival opportunities for the local plum resources, representing a virtuous cycle (Figure 5). For example, in the traditional ecological concept of the Bai people, human beings are part of nature. To destroy nature is to destroy the foundation on which human survival and development depend. Sooner or later, people will usher in nature’s revenge. Therefore, the Bai people’s concept of respecting flowers and plants and conserving the environment is almost deeply ingrained, and a series of customs and habits have also been formed to conserve trees, water, and animals by closing mountains and afforestation, such as planting willow festivals, offering sacrifices to mountain gods, and offering sacrifices to dragon ponds. In addition, through investigation, we found that Eryuan also included the protection of the environment, mountain, forest, and water sources in the custom laws and regulations of various villages and townships. These traditional ecological concepts, customs, and village regulations have subtly protected the local wild plum resources.

Over time, the Bai people’s TMCC has been beneficial for the local biodiversity conservation, indirectly maintaining the stability of the local ecosystem, promoting the sustainable use of plum resources, and boosting the local economic development.

### 3.7. Full Excavation of the Value of Plum

In order to maximize the value of plums, the Eryuan County Government organizes the “Plum Blossom Culture Fair (PBCF)” in February every year. The PBCF adheres to the concept of “ecological priority and green development”. The fair actively promotes the GI of Eryuan Plums, deeply explores the essence of Chinese *mei* plum culture, develops a business brand for the integrated development of Eryuan County’s plum fruit industry and agricultural, cultural, and tourism industry, and continuously expands Eryuan’s external influence.

Relying on the opportunity of the PBCF, the e-commerce center of Eryuan County shows all varieties of plum products to the public through live broadcasting and invites famous local craftsmen to show the traditional production technology of all kinds of plum products, so as to promote the brand of Eryuan plum. Local ethnic groups display traditional arts at the PBCF to publicize traditional ethnic culture, such as playing traditional musical instruments, singing traditional songs, and performing traditional dances. In addition, the public can also enjoy the scenery of plum blossoms, draw or photograph plum blossoms, and participate in other creative works. The PBCF is also broadcast live by Dali Radio and Television Station. Thus, people can appreciate the charm of Eryuan plums and the traditional culture of local ethnic minorities by watching the live broadcast.

While popularizing the Eryuan plum and boosting economic growth, the PBCF also enhances the happiness of the local people. The development of the plum industry in Eryuan has embraced a new situation combining of agricultural development, rural harmony, and farmers’ happiness. Studies have shown that the experiences and practices of one community can be useful to another [63,64,65,66]. The development model of the plum industry in Eryuan may represent a successful case study for other communities around the world.

### 3.8. Local Products and Biodiversity Conservation

Animals, plants, microorganisms, and even landscape ecosystems are adapted to the accumulation of traditional knowledge and practical experience of indigenous communities. This knowledge and these experiences depend on social and environmental conditions according to the nature of local products [67]. It is the constant combination of these factors that underpins and contributes to different levels of biological complexity. Biodiversity, including a range of organisms, their genetics, and their habitats, cannot survive without the practices and knowledge developed by the societies that create, maintain, and protect them. The protection of native products or geographical indication products (GI products) can prompt consideration or even restoration of this cultural biodiversity [68].

Therefore, the protection of local products or GI products implies the need to conserve local ecosystems, animals and plants (landraces, species, subspecies), microbes, and landscapes at different levels. It is also a way of preserving shared knowledge and practice in a formal way. Given that most local products with an appellation of origin are produced in extensive systems linked to local practices and biodiversity, the protection of local products will attract more attention [69,70]. Our case study on traditional *Meizi*-consuming culture in Dali shows the power and vitality of the biocultural diversity of plums. In turn, the sustainable use of plum resources has greatly improved the conservation of biodiversity in Eryuan.

In addition, the internationalization of trade, associated with the free movement of goods, has produced increasingly stringent standards and enforced compliance. These standards are designed in the context of industrial production, with little regard for the characteristics associated with small production units and local products [71]. In order for local products to survive, it is also important to consider criteria appropriate to their characteristics and the situation of the small production unit. Otherwise, in some cases, these local products will disappear, with unpredictable consequences for the biodiversity associated with them [72,73]. Various products made from *Meizi* in Eryuan possess remarkable market value. However, further development (especially across the larger international market) of plum will not be possible without appropriate standards, including those related to biocultural diversity conservation.

## 4. Conclusions

Our research presented a successful case that traditional knowledge related to local products or GI products has important value in biocultural diversity conservation. The ethnobotanical results of this study clearly show that the TMCC of Eryuan Bai people is of great significance for the development of local industry and the conservation of plum genetic resources. A total of 37 species of plants related to TMCC were recorded, belonging to 11 families, among which there were 14 species of plums, including one original species and 13 varieties. By calculating the CFSI values of 37 plants, it was found that *P. mume*, *C. speciosa*, *P. emblica*, *P. salicina*, and *C. cathayensis* had higher CFSI values, indicating that these species play an important role in the Bai people’s lives. There are many types of traditional plum making techniques in the local community, but four production methods of carved plums, preserved plums, perilla-wrapped plums, and stewed plums are the most famous and widely used in industrialization. The Eryuan plum industry not only attaches importance to the economic value of plums, but also contributes ecological value in the process of development. While improving the livelihood of plum farmers and promoting local economic development, the Eryuan plum industry also objectively protects the diversity of local plum resources and maintains the stability of the ecosystem. This development model of Eryuan has achieved a win–win situation in traditional culture, ecological environment, and economic industry, providing an important reference point for rural development in other similar communities.

## Figures and Tables

**Figure 1 biology-11-00832-f001:**
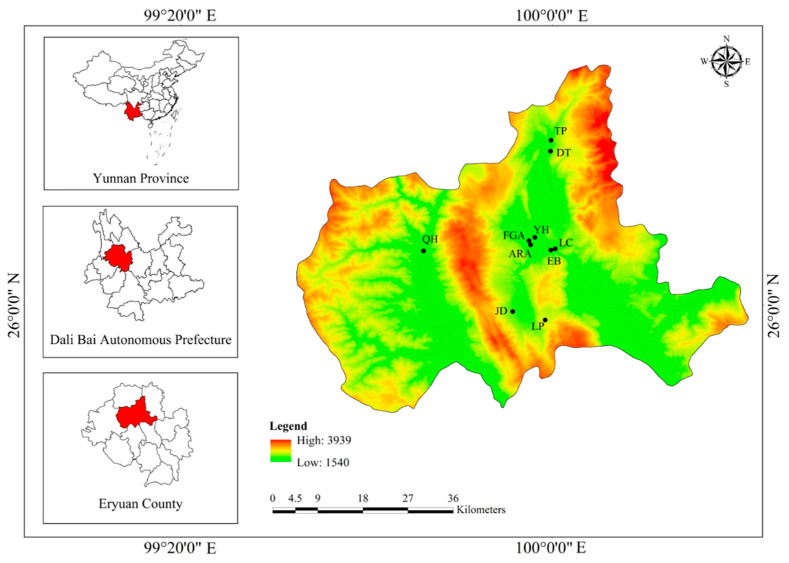
Sketch map of the study area in Eryuan, Dali, Northwest Yunnan, China.

**Figure 2 biology-11-00832-f002:**
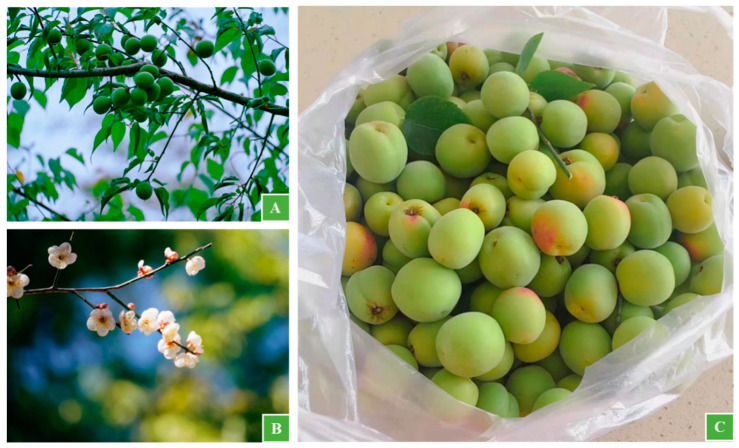
*Mume* plums in the Bai yards ((**A**): plum trees; (**B**): plum blossom; (**C**): plum fruits).

**Figure 3 biology-11-00832-f003:**
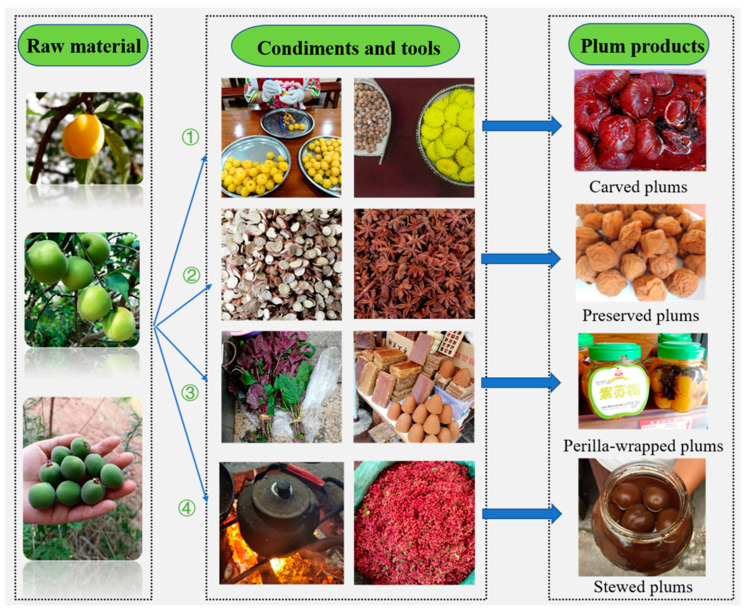
Four famous plum products processed in Eryuan County.

**Figure 4 biology-11-00832-f004:**
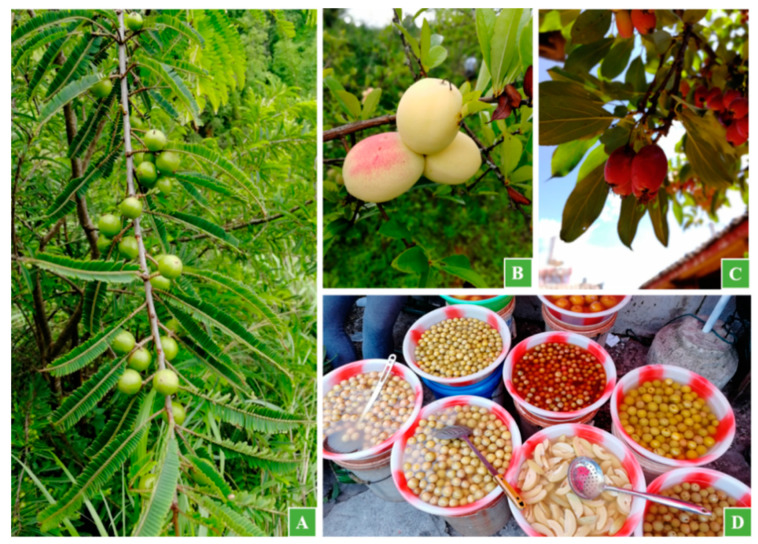
Plants used as substitutes for plums for their similar tastes ((**A**): *Phyllanthus emblica* L.; (**B**): *Chaenomeles cathayensis* (Hemsl.) Schneid.; (**C**): *Malus prunifolia* (Willd.) Borkh.; (**D**): delicacies made from the fruits of these plants).

**Figure 5 biology-11-00832-f005:**
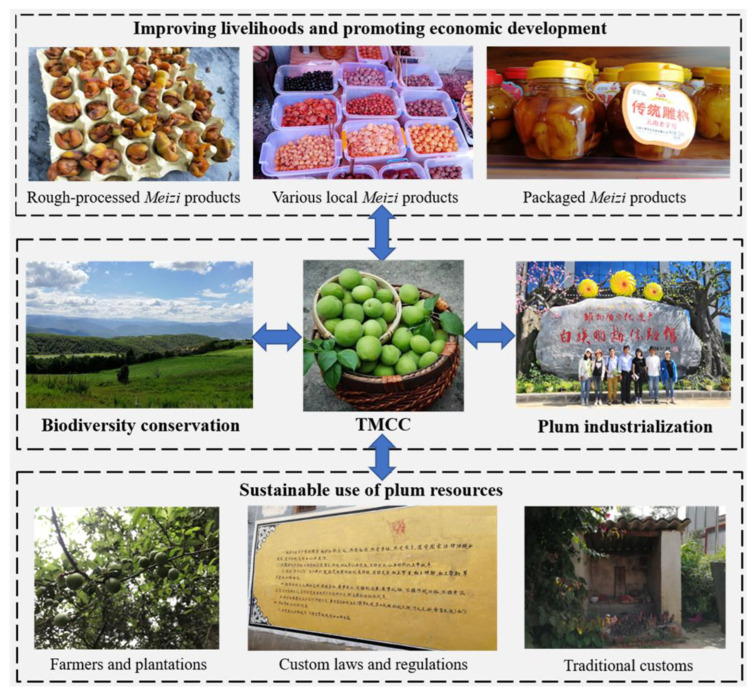
The relationships among the Bai TMCC, local biocultural diversity conservation, and economic development.

**Table 1 biology-11-00832-t001:** Questions used for semi-structured interviews.

Survey Object	Questions
Villager	Do you plant plum in your house? How many trees have you planted?
	Do you like plum? Why?
	How many ways are there to eat plums? How do you make them?
	What are the benefits of eating plum?
	Do you sell plums (or after processing)?
	Is plum important to you? Why?
	Do you know any folk stories about plum?
Market seller	How many plum-related products do you sell?
	Which plum product is the most popular?
	Which plum product is the most expensive?
Enterprise employees	Do you have your own plum planting base? Where?
	How many plum-related products are there? What are the advantages?
	How many local employees are there?
Government agent	Where are the local plums grown?
	How many plum varieties are there?
	Does plum have a large proportion in agriculture?
	Is there any support for relevant enterprises?

**Table 2 biology-11-00832-t002:** The classification and assignment of each index.

Index	Classification	Assignment
AI	Very common	4.0
	Common	3.0
	General	2.0
	Uncommon	1.0
AI correction index	Prevalent	0. 0
	Some places	−0.5
	A certain place	−0.1
FUI	More than 1 time per week	5.0
	1 time per week	4.0
	1 time per month	3.0
	More than 1 time per year but less than 1 time per month	2.0
	1 time per year	1.0
	Unused in the past 30 years	0.5
PUI	Leaf	1.5
	Root	1.5
	Fruit (peel)	1.5
	Seed	1.0
	Bark	1.0
	Flower	0.75
MFFI	Raw eating	1.5
	Cold dishes	1.5
	Water boiling	1.0
	Stewing	1.0
	Seasoning	1.0
	Processing with honey	0.5
	Soaking with wine	0.5
TSAI	Excellent	10.0
	Very good	9.0
	Good	7.5
	Ordinary	6.5
	Poor	5.5
FMRI	Very high (used as medicine)	5.0
	High (as medicine to treat certain diseases)	4.0
	Medium to high (very healthy food)	3.0
	Moderate to low (healthy food, unknown efficacy)	2.0
	Unknown	1.0

**Table 3 biology-11-00832-t003:** Plants related to TMCC used by the Bai people.

Chinese Name	Family	Scientific Name	Local Name	Used Part	Used Category	Used Method	Medicinal Function	No. of Voucher
牛奶子	Elaeagnaceae	*Elaeagnus umbellat**a* Thunb.	Merxu	Fruit	Fru, Sn	RE, CD	PSST	ERY106
甘草	Fabaceae	*Glycyrrhiza uralensis* Fisch.	Gancao	Root	C	Se, WB	TSP, EI	-
紫苏	Lamiaceae	*Perilla frutescens* (L.) Britt.	Zisu	Leaf	C, V, FT	CD, Se	TCH, RC	ERY004
肉桂	Lauraceae	*Cinnamomum cassia* Presl	Rugui	Bark	C, FT	Se	TSP	-
云南杨梅	Myricaceae	*Myrica nana* A. Cheval.	Liwu	Fruit	Fru, Sn	RE, Se, PH, SW	PSST, SAHD	ERY063
丁香蒲桃	Myrtaceae	*Syzygium aromaticum* (L.) Merr. & L.M.Perry	Dingxiangzi	Bud	C	Se	TSP	-
滇橄榄	Phyllanthaceae	*Phyllanthus emblica* L.	Ganglanbei	Fruit	FT, Fru	RE, CD, Se, PH, SW	TST	ERY047
毛叶木瓜	Rosaceae	*Chaenomeles cathayensis* (Hemsl.) Schneid.	Muguer	Fruit	C, FT, Sn, Be	CD, WB, PH	SAHD	ERY041
皱皮木瓜	Rosaceae	*Chaenomeles speciosa* (Sweet) Nakai	Muguer	Fruit	C, FT, Sn, Be	CD, WB, PH	SAHD	ERY005
云南山楂	Rosaceae	*Crataegus scabrifolia* (Franch.) Rehd.	Shanzha	Fruit	FT, Fru, Sn	RE, Se, SW	SAHD	ERY054
云南多依	Rosaceae	*Docynia delavayi* (Franch.) Schneid.	Du yi	Fruit	Sn	CD, Se	SAHD, ICD	ERY014
海棠果	Rosaceae	*Malus prunifolia* (Willd.) Borkh.	Haittal	Fruit	Fru	RE, Se	-	ERY086
绿萼梅	Rosaceae	*Prunus mume* f. *viridicalyx* (Makino) T. Y. Chen	Ji	Fruit	Fru, Sn, Be	RE, CD, Se, PH, St	PSST	-
冰梅	Rosaceae	*Prunus mume* var. *albopl**en**a* Bailey	Ji	Fruit	Fru, Sn, Be	RE, CD, Se, PH, St	PSST	-
长梗梅	Rosaceae	*Prunus mume* var. *cernua* (Franch.) Yü et Lu	Ji	Fruit	Fru, Sn, Be	RE, CD, Se, PH, St	PSST	ERY143
毛梅	Rosaceae	*Prunus mume* var. *goethartiana* Koehne	Ji	Fruit	Fru, Sn, Be	RE, CD, Se, PH, St	PSST	-
早梅	Rosaceae	*Prunus mume* var. *macrocarpa* Mak.	Ji	Fruit	Fru, Sn, Be	RE, CD, Se, PH, St	PSST	-
照水梅	Rosaceae	*Prunus mume* var. *pendula* Sieb.	Ji	Fruit	Fru, Sn, Be	RE, CD, Se, PH, St	PSST	-
品字梅	Rosaceae	*Prunus mume* var. *pleiocarpa* Maxim.	Ji	Fruit	Fru, Sn, Be	RE, CD, Se, PH, St	PSST	-
紫梅	Rosaceae	*Prunus mume* var. *purpurea* Mak.	Ji	Fruit	Fru, Sn, Be	RE, CD, Se, PH, St	PSST	-
梅	Rosaceae	*Prunus mume* Sieb. & Zucc.	Ji	Fruit	Fru, Sn, Be	RE, CD, Se, PH, St	PSST	ERY071
白梅	Rosaceae	*Prunus mume* var. *alba* Rehd.	Ji	Fruit	Fru, Sn, Be	RE, CD, Se, PH, St	PSST	-
红梅	Rosaceae	*Prunus mume* var. *alplandii* Rehd.	Ji	Fruit	Fru, Sn, Be	RE, CD, Se, PH, St	PSST	-
杏梅	Rosaceae	*Prunus mume* var. *bung* Mak.	Ji	Fruit	Fru, Sn, Be	RE, CD, Se, PH, St	PSST	-
细梅	Rosaceae	*Prunus mume* var. *cryptopetala* Mak.	Ji	Fruit	Fru, Sn, Be	RE, CD, Se, PH, St	PSST	-
厚叶梅	Rosaceae	*Prunus mume* var. *pallescens* (Franch.) Yü et Lu	Ji	Fruit	Fru, Sn, Be	RE, CD, Se, PH, St	PSST	-
李子	Rosaceae	*Prunus salicina* Lindl.	He	Fruit	Fru, Sn	RE, CD, SW	PSST	ERY032
川梨	Rosaceae	*Pyrus pashia* Buch.-Ham. ex D. Don	Xuli	Fruit	Sn	Se	-	ERY027
沙梨	Rosaceae	*Pyrus pyrifolia* (Burm. F.) Nakai	Xuli	Fruit	FT, Fru	RE, Se, WB	MDL, RC	ERY007
刺梨	Rosaceae	*Rosa roxburghii* Tratt.	Cilingguo	Fruit	FT, Fru	RE, Se, PH, SW	SAHD	ERY024
玫瑰	Rosaceae	*Rosa rugosa* Thunb.	Meigui	Flower	C, FT	Se	ICD	ERY051
柠檬	Rutaceae	*Citrus limon* (L.) Osbeck	Zeng	Fruit	C	Se	PSST	ERY043
橘	Rutaceae	*Citrus reticulata* Blanco	Zuzi	Peel	C, FT	Se	SAHD, RP	ERY051
花椒	Rutaceae	*Zanthoxylum bungeanum* Maxim.	Su	Seed	C, FT	Se, St	DWD, RC	ERY049
八角	Schisandraceae	*Illicium verum* Hook. f.	Weixiaozi	Seed	C, FT	Se	SAHD	ERY141
辣椒	Solanaceae	*Capsicum annuum* L.	Lazi	Fruit	C, FT	CD, Se	SAHD	ERY118
枸杞	Solanaceae	*Lycium chinense* Miller	Gouqi	Fruit	C, FT	Se	TD	-

Species in the inventory are arranged alphabetically by the family name. Local names are written using Chinese pinyin. Use category: C, condiment; V, vegetable; FT, food therapy; Fru, fruit; Sn, snack; Be, beverage. Used method: RE, raw eating; CD, cold dishes; Se, seasoning; WB, water boiling; PH, processing with honey; SW, soaking with wine; St, stewing. Medicinal function: PSST, produce saliva and slake thirst; SAHD, stimulate appetite and help digestion; TSP, treat stomach pain; EI, eliminate inflammation; RP, reduce phlegm; TCH, treat cold and headache; RC, relieve cough; TST, treat sore throat; TD, tonify deficiency; ICD, invigorate circulation of blood; MDL, moisten dry lungs; DWD, dispel wind dampness; -, without definite food–medicinal value.

**Table 4 biology-11-00832-t004:** Ranking of Cultural Food Significance Index (CFSI) of traditional botanical knowledge associated with plums in the Bai communities.

Species	FQI	AI	FUI	PUI	MFFI	TSAI	FMRI	CFSI
*Prunus mume* Sieb. & Zucc.	68	5	5	1.5	5	7.5	5	4781.25
*Chaenomeles speciosa* (Sweet) Nakai	64	5	5	1.5	5	7.5	5	4500.00
*Phyllanthus emblica* L.	64	5	4	1.5	5	7.5	5	3600.00
*Prunus salicina* Lindl.	57	5	4	1.5	5	7.5	5	3206.25
*Chaenomeles cathayensis* (Hemsl.) Schneid.	59	4	4	1.5	5	7.5	5	2655.00
*Pyrus pyrifolia* (Burm. f.) Nakai	52	4	3	1.5	3	9	5	1263.60
*Perilla frutescens* (L.) Britt.	32	5	4	1.5	2.5	9	5	1080.00
*Crataegus scabrifolia* (Franch.) Rehd.	43	4	3	1.5	3	9	5	1044.90
*Rosa roxburghii* Tratt.	46	3	3	1.5	3.5	7.5	5	815.06
*Capsicum annuum* L.	34	4	5	1.5	1.5	6.5	5	497.25
*Docynia indica* (Wall.) Dcne.	38	3	3	1.5	2.5	7.5	4	384.75
*Elaeagnus umbellat**a* Thunb.	53	3	3	1.5	3	7.5	2	321.98
*Myrica nana* A. Cheval.	19	4	3	1.5	2.5	9	3	230.85
*Zanthoxylum bungeanum* Maxim.	31	3	5	1	1.5	6.5	5	226.69
*Citrus reticulata* Blanco	37	3	3	1.5	1	9	5	224.78
*Prunus mume* var. *cernua* (Franc.) Yü et Lu	23	2	2	1.5	5	7.5	3	155.25
*Citrus limon* (L.) Osbeck	29	3	3	1.5	1	6.5	5	127.24
*Prunus mume* var. *pendula* Sieb.	21	2	2	1.5	5	6.5	3	122.85
*Pyrus pashia* Buch.-Ham. ex D. Don	41	3	2	1.5	5	5.5	1	101.48
*Prunus mume* var. *albopl**en**a* Bailey	17	2	2	1.5	5	6.5	3	99.45
*Prunus mume* var. *goethartiana* Koehne	17	2	2	1.5	5	6.5	3	99.45
*Prunus mume* var. *purpurea* Mak.	17	2	2	1.5	5	6.5	3	99.45
*Prunus mume* f. *viridicalyx* (Mak.) T. Y. Chen	17	2	2	1.5	5	6.5	3	99.45
*Prunus mume* var. *bung* Mak.	17	2	2	1.5	5	6.5	3	99.45
*Prunus mume* var. *cryptopetala* Mak.	17	2	2	1.5	5	6.5	3	99.45
*Prunus mume* var. *pleiocarpa* Maxim.	17	2	2	1.5	5	6.5	3	99.45
*Prunus mume* var. *alba* Rehd.	17	2	2	1.5	5	6.5	3	99.45
*Prunus mume* var. *alplandii* Rehd.	17	2	2	1.5	5	6.5	3	99.45
*Prunus mume* var. *pallescens* (Franch.) Yü et Lu	17	2	2	1.5	5	6.5	3	99.45
*Lycium chinense* Miller	23	3	2	1.5	1	7.5	5	77.63
*Rosa rugosa* Thunb.	19	4	4	0.75	1	6.5	5	74.10
*Illicium verum* Hook. f.	15	3	4	1	1	6.5	5	58.50
*Prunus mume* var. *macrocarpa* Mak.	17	2	2	1.5	2.5	6.5	3	49.73
*Glycyrrhiza uralensis* Fisch.	21	1	2	1.5	2	5.5	5	34.65
*Cinnamomum cassia* Presl	11	1	3	1	1	6.5	5	10.73
*Malus prunifolia* (Willd.) Borkh.	14	2	1	1.5	2.5	5	1	5.25
*Syzygium aromaticum* (L.) Merr. & L.M. Perry	7	1	2	1	1	6.5	4	3.64

## Data Availability

The data presented in this study are available on request from the corresponding authors.

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
