# Peer review of "Meizi-Consuming Culture That Fostered the Sustainable Use of Plum Resources in Dali of China: An Ethnobotanical Study"

_biology, 2022, doi:10.3390/biology11060832_

Round 1
Reviewer 1 Report
Reviewer report
Manuscript title: Plum-consuming Culture That Fostered the Sustainable Use of
Plum Resources in Dali of China: An Ethnobotanical Study
This work is very interesting, and important because it intervenes to enhance the traditional use (food and medicinal) and processing methods of plum fruits by the Bai population of Dali in China.
I find that this work is good. So, I have no comments, except one concerning the tables (1, 2 and 3). If it is possible to separate the rows of tables by lines in order to let us understand the information presented.
For table 1: to know exactly which questions are asked on which category?
For table 2: to separate the different index
For Table 3: there is overlap especially in the columns of used category, used method and medicinal function.
And the second concerning
Line 209: I think that it is better to change “eliminate inflammation” to “anti-inflammatory effect” or “treat inflammation” or “reduce inflammation”
Line 221 to line 231: the scientific names of the species mentioned in this paragraph must be written in italics
Line 278 to line 297: idem as previous remark
Reviewer 2 Report
I have read the paper with great interest. I think you should avoid using the English phytonym plum in the title since it can be confused with the more common P. domestica. Instead use the English name Chinese plume or Mume plume. In the summary and abstract you can use only the scientifc name Prunus mume, and when it is mentioned first time add the author names Prunus mume (Siebold) Siebiold & Zucc. Biology is a scientific journal, therefore full scientific names with author should be standard.
In the text there are many scientific names which should be given in italic!
You could some basic information about the Bai people, for instance their number (according to 2010 census 1.9 Millions) and other economic activities (fishing), that their lanuage belong to Sino-Tibetan family etc.
Are all local names given in table 3 in Bai language or in Mandarin. Perilla is for instance known as zisu in Mandarin, Maybe it is a loan word in Bai.
p. 10 what is green manure plants? Plants grown to be used as manure or weed? Please clarify.
Reviewer 3 Report
The manuscript entitled „Plum-consuming Culture That Fostered the Sustainable Use of Plum Resources in Dali of China: An Ethnobotanical Study” by Yanxiao Fan is an interesting study aiming to explain the relationship among the Bai’s traditional plum-consuming culture and conservation of local plum resources as well as the development of plum industry in Eryuan. The paper is generally well written. I recommend that authors perform a review in terms of minor spell checks and format, for example the latin name of plant/plum species should be written with italic font (for example see page 8, lines 223-231; page 10, line 255; page 11, lines 279-282 etc). My personal opinion is that authors should perform a statistical analysis to the data recorded and also some correlations between the indices followed by their study; I also think that an extensive critical discussion of the results in relation to the existing literature in the field of their research could improve the manuscript.
